# Which Data Attributes Stimulate Math and Code Reasoning? An Investigation via Influence Functions

**Siqi Kou, Qingyuan Tian, Hanwen Xu, Zihao Zeng, and Zhijie Deng**[*]
Shanghai Jiao Tong University
{happy-karry, qy.tian, dctnorin, zengzihao, zhijied}@sjtu.edu.cn

## Abstract

Large language models (LLMs) have demonstrated remarkable reasoning capabilities in math and coding, often bolstered by post-training on the chain-of-thoughts (CoTs) generated by stronger models. However, existing strategies for curating such training data predominantly rely on heuristics, limiting generalizability and failing to capture subtleties underlying in data. To address these limitations, we leverage influence functions to systematically attribute LLMs' reasoning ability on math and coding to individual training examples, sequences, and tokens, enabling deeper insights into effective data characteristics. Our **Influence-based Reasoning Attribution (Infra)** uncovers nontrivial cross-domain effects across math and coding tasks: high-difficulty math examples improve both math and code reasoning, while low-difficulty code tasks most effectively benefit code reasoning. Based on these findings, we introduce a simple yet effective dataset reweighting strategy by flipping task difficulty, which doubles AIME24 accuracy from 10% to 20% and boosts LiveCodeBench accuracy from 33.8% to 35.3% for Qwen2.5-7B-Instruct. Moreover, our fine-grained attribution reveals that the sequence-level exploratory behaviors enhance reasoning performance in both math and code, and the token-level influence patterns are distinct for math and code reasoning: the former prefers natural language logic connectors and the latter emphasizes structural syntax.

## 1 Introduction

Large language models (LLMs) for reasoning, with OpenAI-o1 [15] and DeepSeek-R1 [9] as popular examples, have shown great promise in solving complex math and coding problems. Recently, the community has witnessed the prevalence of reproducing such reasoning capacities on open-source small- to medium-sized LLMs [19, 5, 29]. An initial stage of the solutions often involves post-training the model on some chain-of-thought (CoT) reasoning traces curated by leading models (e.g., R1) for diverse problems [34, 23, 24, 37, 13, 25]. As a data-centric paradigm, the core research question here is: *which attributes of the training data are effective in stimulating reasoning capabilities?*

Pioneering studies addressing this question predominantly adopt heuristic approaches. Typically, they first establish quantitative data quality metrics based on human expertise or empirical preferences, then selectively retain high-quality data for model training to cultivate robust reasoning capabilities with minimal data inputs. For example, s1K [24] filters 1k (question, answer) pairs with well-structured formatting, longer CoT length, and broader domain coverage from an initial pool of 59k data for training math reasoning LLMs. Similarly, LIMO [37] suggests incorporating more challenging math questions with complex reasoning chains to enable better math reasoning.

Beyond focusing exclusively on math, Sky-T1 [25] targets competitive reasoning performance across both math and coding tasks. It notices that the naive incorporation of code data from APPS [10]

---

[*]Corresponding author.

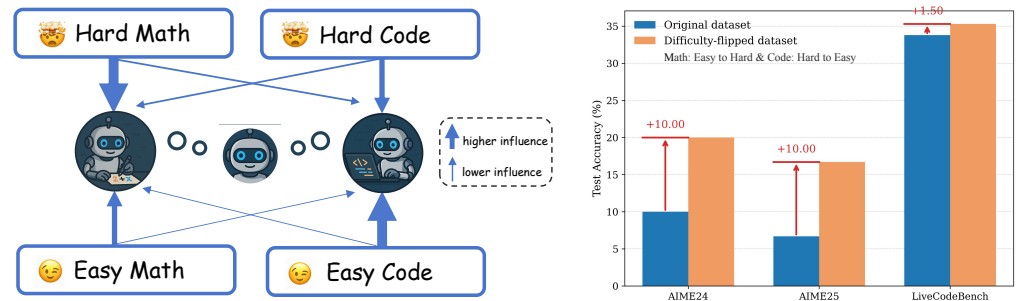

Figure 1: An illustration of our key findings towards the question: *Which attributes of training data effectively stimulate reasoning capabilities?* Mixing challenging math problems with easier coding tasks leads to the highest influence scores for mathematical and coding reasoning (*left*). Guided by this insight, we curate an improved dataset and observe enhanced performance (*right*).

degrades math performance and advocates mitigating this by introducing more difficult math questions and code tasks for training. Nevertheless, the underlying mechanism of such cross-domain influence remains underexplored. Furthermore, these heuristic strategies suffer from unreliable generalization to other reasoning scenarios and cannot clearly explain how some fine-grained reasoning patterns in the training data (e.g., verification, backtracking, etc.) affect the learned models.

To bridge the gap, we leverage influence functions [17]—a classical technique for tracing the impacts of individual training data on model behavior—to systematically identify which training examples, along with their internal patterns and tokens, most significantly enhance the reasoning capabilities on math and coding tasks. Following previous works on influence functions for LLMs [8, 30], we define an easy-to-implement and cost-effective influence function for reasoning-oriented supervised fine-tuning (SFT). We further extend the instance-wise influence function to more fine-grained variants at the sequence and token levels for an in-depth data attribution. We dub our approach as Infra.

We begin by investigating cross-domain influence in basic math and code reasoning scenarios without long CoT. To this end, we fine-tune LLaMA3-8B-Base [7] on a mixture of MetaMathQA [39] and OSS-Instruct [33] datasets and compute the influence function on the accuracy of GSM8k [3] and MBPP [10]. We rank all training data by their influence scores and find that, *while in-domain data yield the highest scores as expected, cross-domain data also contribute nontrivially.* Furthermore, aggregating these scores by category and difficulty reveals that symbolic math examples and high-difficulty math problems are particularly effective in improving code reasoning.

Extending Infra to complex long CoT reasoning, we fine-tune Qwen2.5-7B-Instruct [36] on Bespoke-Stratos-17k[2] dataset and measure influence using AIME, MATH500 [11], and LiveCodeBench [16] benchmarks. Consistent with earlier findings, we observe cross-domain gains, with harder math problems better helping code reasoning. Going a step further, we find that *both high-difficulty math and code examples are more influential on math reasoning, whereas low-difficulty code tasks contribute most significantly to code reasoning (see Figure 1).* Motivated by these insights, we flip easy math problems as hard and hard code tasks as easy in the training data. This reweighted dataset doubles AIME accuracy and improves LiveCodeBench accuracy from 33.8% to 35.3%.

Furthermore, we perform attribution at sequence and token levels in long CoT. Sequence-level attribution shows that *the exploration behavior of seeking alternative approaches after reaching correctness (refer to Figure 6), which is common in long CoTs, improves both math and code reasoning performance.* Despite being seen as overthinking [2, 32], our studies suggest it is advantageous. Besides, we observe *distinct token-level influence patterns for math and code reasoning.* In math, the most influential tokens are natural language with logical connectors, whereas code CoTs rely more on syntax markers. This divergence explains why easier code problems with clearer structural solutions benefit code reasoning when combined with math CoT that already provides logical skills.

## 2   Related Work

**LLM reasoning.** Reasoning is a cognitive process that involves using evidence, arguments, and logic to arrive at conclusions or make judgments. It is widely regarded as a foundational element of

---

[2]`https://huggingface.co/datasets/bespokelabs/Bespoke-Stratos-17k`

advanced Large Language Models (LLMs) and an essential milestone on the path toward Artificial General Intelligence (AGI) [12, 28, 29, 15, 35]. A very recent approach to achieve reasoning capacity in LLMs is through post-training, such as OpenAI-o1 [15], and Deepseek-R1 [9], which expose the model to large-scale curated reasoning examples after the initial pretraining phase to refine its inferential abilities [18]. These reasoning datasets predominantly fall into two categories: (1) Mathematical reasoning: In earlier work, the construction of high-quality mathematical datasets primarily relied on increasing the quantity of problems and enhancing their difficulty levels [20, 40]. Nevertheless, LIMO dataset [37] demonstrated that complex reasoning capabilities can be elicited through surprisingly small datasets (hundreds of examples). In addition, some researchers also opted to distill high-quality reasoning data from strong LLMs [25], leveraging their outputs to construct more targeted and informative training sets for enhancing reasoning performance in weak LLMs. (2) Code generation: As a highly structured and formalized type of data, code has a non-negligible impact on the development of reasoning abilities in large language models. Beyond simply testing LLMs on newly coding test cases [16], many efforts have focused on investigating how and when code data influences the development of reasoning abilities in language models [41, 21]. In our work, we consider mathematical capacity and coding ability as two distinct manifestations of advanced reasoning, and we aim to analyze and understand the interactions between these capabilities to gain deeper insights into the underlying mechanisms of LLM reasoning.

**Data attribution and influence functions.** Training Data Attribution (TDA) methods seek to interpret a model's predictions by analyzing the particular training instances that contributed to shaping its learned representations. Most modern TDA methods can broadly be divided into two categories: retraining-based methods [22, 31, 14] and gradient-based methods [38, 27, 17]. However, applying traditional data attribution methods to large language models has remained a significant challenge, primarily due to issues of computational tractability and the sheer scale of model parameters. Nonetheless, there are several works successfully apply data attribution on LLMs by influence function. Researchers in Anthropic adapt EK-FAC influence functions to large-scale Transformers, by which they figured out what kind of pretraining data influences completions of models up to 50B parameters [8]. More specifically, for reasoning capabilities, studies have shown that code data encountered during the pretraining-phase plays a critical role in the development of mathematical reasoning abilities in language models. [30]. In this work, we extend similar methodological approaches by employing influence functions to attribute the development of reasoning capabilities during the supervised fine-tuning (SFT) phase, with a particular focus on analyzing the interplay between code and mathematical data.

## 3 Methodology

This section reviews the basics of influence functions [17, 8] and presents Infra, our adaptation for attributing LLM reasoning performance on math and code problems. In particular, we compute instance-level influence scores using a mean log-likelihood proxy, and further shift to the sequence and token levels to uncover how specific reasoning steps and tokens shape model behavior.

### 3.1 Preliminary: Influence Functions

Given a model parameterized by $\boldsymbol{\theta}$ and trained on a dataset $\mathcal{D}_{\text{train}} = \{z_i\}_{i=1}^N$, influence functions [17] estimate the influence of a training point $z_m$ on $\boldsymbol{\theta}$ (or a function thereof) without retraining the model. Specifically, it is measured by computing the change in $\boldsymbol{\theta}$ if $z_m$ is upweighted by an infinitesimal amount $\epsilon$. This perturbation can be formalized as the response function[3]:

$$\boldsymbol{\theta}(\epsilon) = \underset{\boldsymbol{\theta} \in \mathbb{R}^D}{\arg\min} \mathcal{J}\left(\boldsymbol{\theta}, \mathcal{D}_{\text{train}}, \epsilon\right) = \underset{\boldsymbol{\theta} \in \mathbb{R}^D}{\arg\min} \frac{1}{N} \sum_{i=1}^N \mathcal{L}\left(z_i, \boldsymbol{\theta}\right) + \epsilon \mathcal{L}\left(z_m, \boldsymbol{\theta}\right), \quad (1)$$

where $\mathcal{L}(\cdot)$ is the training loss. The influence of $z_m$ on $\boldsymbol{\theta}$ is then defined as the first-order Taylor approximation to the response function around $\epsilon = 0$ and can be computed using the implicit theorem:

$$\mathcal{I}_{\boldsymbol{\theta}}(z_m) = \left.\frac{d\boldsymbol{\theta}}{d\epsilon}\right|_{\epsilon=0} = -\mathbf{H}^{-1} \nabla_{\boldsymbol{\theta}} \mathcal{L}\left(z_m, \boldsymbol{\theta}\right), \quad (2)$$

---

[3]For simplicity, we show the response function for optimal parameters. For non-converged or non-convex models, the actual response function is the Proximal Bregman response function (refer to [8] for details).

where $\mathbf{H} = \nabla_{\boldsymbol{\theta}}^2 \mathcal{J}(\boldsymbol{\theta}, \mathcal{D}_{\text{train}})$ is the Hessian of the cost function. Direct interpretation of $\mathcal{I}_{\boldsymbol{\theta}}(z_m)$ can be difficult due to its high dimensionality, so it is common to instead compute the influence of $z_m$ on a scalar-valued function of the parameters $f(\boldsymbol{\theta})$. Using the chain rule for derivatives, this influence admits the closed-form:

$$\mathcal{I}_f(z_m) = \left.\frac{df(\boldsymbol{\theta})}{d\epsilon}\right|_{\epsilon=0} = -\nabla_{\boldsymbol{\theta}} f(\boldsymbol{\theta})^T \mathbf{H}^{-1} \nabla_{\boldsymbol{\theta}} \mathcal{L}(z_m, \boldsymbol{\theta}). \tag{3}$$

A complete derivation of Equation 3 is delayed to Appendix A. Consequently, $f(\boldsymbol{\theta})$ is expected to increase after upweighting the sample $z_m$ and then retraining the model if $\mathcal{I}_f(z_m) > 0$, as

$$f(\boldsymbol{\theta}(\epsilon)) - f(\boldsymbol{\theta}) \approx \mathcal{I}_f(z_m)\epsilon = -\nabla_{\boldsymbol{\theta}} f(\boldsymbol{\theta})^T \mathbf{H}^{-1} \nabla_{\boldsymbol{\theta}} \mathcal{L}(z_m, \boldsymbol{\theta}) \epsilon. \tag{4}$$

For transformer-based LLMs with billions of parameters, the above $\mathbf{H}$ is intractable. To address this, Grosse et al. [8] propose to approximate $\mathbf{H}$ using the Eigenvalue-Corrected Kronecker-Factored Approximate Curvature (EK-FAC) method [6], which introduces simplifying assumptions such as layer-wise independence and restricts computation only to the MLP parameters within the model. Given the effectiveness of such a strategy, we also employ it to effectively estimate influence scores.

## 3.2 Attributing LLM Reasoning to Training Data via Influence Functions

We now introduce Infra, our adaptation of influence functions to attribute LLM reasoning on challenging math and code tasks. As mentioned, our setting is mainly an SFT process with CoTs generated by a stronger model to improve the reasoning abilities of the LLM at hand. We are interested in identifying the most influential training data to improve model performance. Since task accuracy is non-differentiable with respect to $\boldsymbol{\theta}$, we instead adopt a smooth surrogate: the mean log-likelihood over a set of correctly answered examples. Let $\mathcal{D}_{\text{correct}} = \{(\boldsymbol{x}_i, \boldsymbol{y}_i)\}_{i=1}^n$ denote a collection of problems $\boldsymbol{x}_i$ paired with correct answers $\boldsymbol{y}_i$, we define the surrogate objective as:

$$f(\boldsymbol{\theta}) = \frac{1}{n} \sum_{i=1}^n \log p(\boldsymbol{y}_i|\boldsymbol{x}_i; \boldsymbol{\theta}), \tag{5}$$

where $n$ is the size of $\mathcal{D}_{\text{correct}}$. The robustness of $\mathcal{D}_{\text{correct}}$ against variation is ablated in Appendix C.

**Instance-level influence scores.** Plugging Equation 5 into Equation 3 yields the instance-level influence score assigned to each SFT training example $z_m$ reflecting its effect on $f(\boldsymbol{\theta})$. Consistent with [8], we restrict our focus to positively influential data, which refers to data points that yield an increase in the log-likelihood of correct answers and thus more effectively enhances the model's reasoning performance.

**Sequence-level influence scores.** Reasoning traces of recent models often exhibit sequence-level cognitive behaviors, such as verification or exploration (refer to Figure 6). To attribute the contribution of an individual sentence $\boldsymbol{y}$ in $z_m$, we employ a simple counterfactual tactic: we remove $\boldsymbol{y}$ from the example and measure how the influence scores changes. Let $z_m^{\backslash \boldsymbol{y}}$ denote the input with sentence $\boldsymbol{y}$ erased. Then the sequence-level influence of $\boldsymbol{y}$ is given by

$$\mathcal{I}_f(\boldsymbol{y}) = \mathcal{I}_f(z_m) - \mathcal{I}_f(z_m^{\backslash \boldsymbol{y}}), \tag{6}$$

which isolates the influence of $\boldsymbol{y}$ on the target function $f(\boldsymbol{\theta})$.

**Token-level influence scores.** Tokens that mark critical transitions—such as 'wait'—frequently appear in long CoT. Attributing influence at the token level may therefore help elucidate the underlying mechanisms that guide the model's reasoning. Due to the autoregressive nature of LLMs, the training gradient of a training sequence $z_m$ of length $T$ decomposes as:

$$\nabla_{\boldsymbol{\theta}} \mathcal{L}(z_m, \boldsymbol{\theta}) = \sum_{t=1}^T -\nabla_{\boldsymbol{\theta}} \log p(z_{m,t}|z_{m,<t}, \boldsymbol{\theta}), \tag{7}$$

where $z_{m,t}$ denotes the $t$-th token and $z_{m,<t} = \{z_{m,1}, \ldots, z_{m,t-1}\}$. Plugging this into Equation 3 yields the token-level influence of $z_{m,t}$:[4]

$$\mathcal{I}_f(z_{m,t}) = \nabla_{\boldsymbol{\theta}} f(\boldsymbol{\theta})^T \mathbf{H}^{-1} \nabla_{\boldsymbol{\theta}} \log p(z_{m,t}|z_{m,<t}, \boldsymbol{\theta}). \tag{8}$$

---

[4]This term captures the influence of $z_{m,t}$ as the output for the model to fit, ignoring its role as input in other cases, for simplicity.

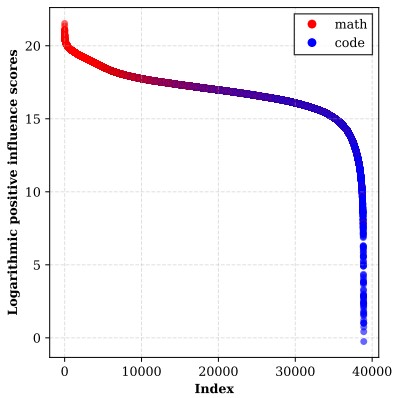
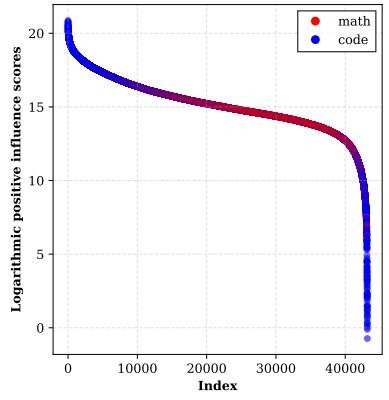

(a) Source of data with top influence scores on math    (b) Source of data with top influence scores on code

Figure 2: Cross-domain influence analysis of LLaMA3-8B-Base fine-tuned on combined Meta-MathQA and OSS-Instruct for math and code performance. The most beneficial examples for math performance predominantly come from the math domain, while code-domain data also contributes non-trivially (left). A similar cross-domain benefit is observed for code performance (right).

## 4 Experiments

We begin by detailing the experimental setup (§4.1), and then present the main findings, progressing from coarse- to fine-grained analyses (§4.2–§4.4).

### 4.1 Experimental Setup

We conduct experiments under two SFT settings and interpret math and code reasoning behaviors using influence functions in both scenarios. All training and influence scores computation are carried out on servers equipped with 8 NVIDIA A100 80GB GPUs.

**Base models trained w/o long CoT.** We fine-tune the Llama3-8B-Base model [7] using a mixed training dataset comprising MetaMathQA-100k [39] and OSS-Instruct-75k [33]. MetaMathQA-100k includes reformulated questions bootstrapped from training splits of GSM8k [3] and MATH [11] paired with brief answers (~100 tokens) generated from GPT-3.5-Turbo [26]. OSS-Instruct-75k provides synthetically generated instructions covering a range of coding tasks. We evaluate the resulting model on the test splits of GSM8k and MBPP [1], filtering correctly answered data to compute influence scores. The MBPP benchmark consists of 1,000 Python programming problems, each comprising a task description and three automated test cases.

**Instruction-tuned models trained w/ long CoT.** We fine-tune the Qwen2.5-7B-Instruct model [36] on the Bespoke-Stratos-17k reasoning dataset[5] (BS-17k), which includes SFT distillation data from DeepSeek-R1 [9], comprising questions, reasoning traces, and answers. We employ the AIME24, AIME25, MATH500, and LiveCodeBench [16] benchmarks to evaluate reasoning performance. AIME is a prestigious high school mathematics competition known for its challenging problems. MATH500 is a subset of 500 problems drawn from the MATH [11] benchmark. LiveCodeBench evaluates LLMs on diverse coding tasks, including self-repair, code execution, and test output prediction, and currently hosts 400 coding problems.

**Influence scores computation.** We estimate the Hessian using EK-FAC on the full SFT training set, truncating sequences to 4096 tokens to reduce memory usage. We set $n = 100$ in Equation 5 by randomly sampling correctly answered math and code examples.

### 4.2 Instance-level Attribution

> **Finding 1:**
>
> *Code data can positively influence math performance, and vice versa.*

---
[5]https://huggingface.co/datasets/bespokelabs/Bespoke-Stratos-17k

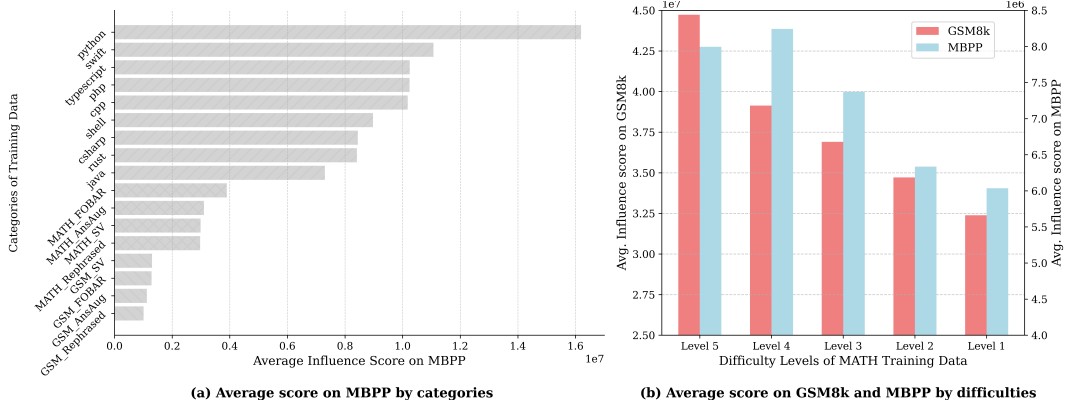

(a) Average score on MBPP by categories      (b) Average score on GSM8k and MBPP by difficulties

Figure 3: Average influence score of the training dataset combining MetaMathQA and OSS-Instruct, evaluated on MBPP and GSM8K performance. Results are grouped by training data category (left) and MATH problem difficulty (right).

To investigate cross-domain influence after fine-tuning LLaMA3-8B-Base on MetaMathQA and OSS-Instruct, we rank training samples based on their positive influence on the mean likelihood of correct answers in math and coding tasks, respectively, and categorize them by domain. As shown in Figure 2 (a), the most influential samples for improving math performance predominantly originate from the math domain. However, influence scores from code-domain data are not narrowly concentrated in the low range (0–10); instead, a substantial number exhibit scores in the 15–20 range, indicating a non-trivial contribution from code to math. A similar pattern of cross-domain benefit is observed in Figure 2 (b). This also holds in long CoT reasoning scenarios as shown in Appendix B.

To investigate how various training data types influence code reasoning, we further aggregate training samples by category and compute average influence scores per category. As illustrated in Figure 3 (a), in-domain Python data yields the highest average influence on MBPP (a benchmark of 1,000 Python problems). Within the math domain, symbolic problem-answer pairs—such as those introducing variables $x$ in FOBAR and SV formats shown in Figure 4—most effectively enhance coding capabilities. Moreover, college-level math questions from the MATH dataset, which utilize LaTeX-based formal expressions, contribute more positively to code performance than simpler, conversational high-school problems from GSM8k. This suggests that, beyond domain relevance, the complexity and formality of the data—especially the use of precise symbolic language—also play a critical role in enabling models to generalize effectively to code reasoning tasks.

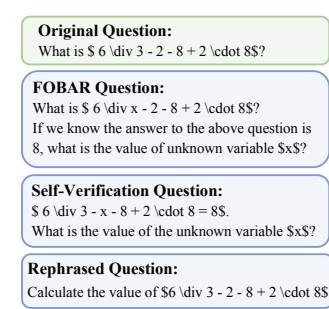

Figure 4: Different types of MATH questions from Meta-MathQA [39] dataset.

> **Finding 2:**
>
> *Challenging math problems exhibit higher influence scores on both math and code reasoning, while simpler code problems more effectively enhance code tasks when combined with math data. The optimal strategy for co-optimizing reasoning across both domains is to mix challenging math problems with easier code tasks.*

To examine how training data difficulty contributes to model performance, we first categorize MATH training data into different difficulty levels and compute the average influence score for each level. As shown in Figure 3(b), higher-difficulty problems (Level 5 and 4) contribute more significantly to performance improvements on GSM8k and MBPP compared to lower-difficulty ones (Level 3, 2, and 1). This may be attributed to the fact that high-difficulty MATH problems induce more complex reasoning chains and thus better transfer logical capabilities to other reasoning-intensive tasks.

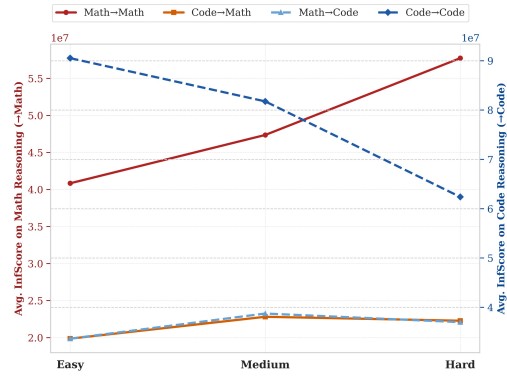
(a) Influence of Math and Code Data on Reasoning Across Difficulty Levels

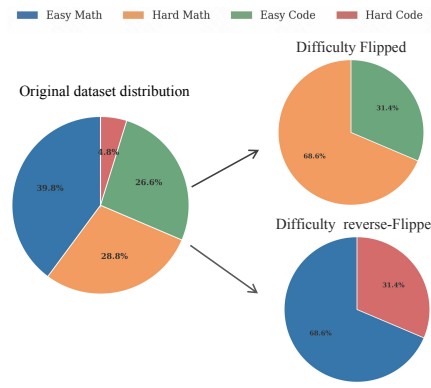
(b) Illustration of Training Dataset Distribution

Figure 5: *Left:* Average influence scores of math and code training data from varying difficulty levels on reasoning performance. For instance, Math→Code denotes the influence of math data on code reasoning tasks. *Right:* Distribution of math and code samples across difficulty levels in the BS17k dataset. The original distribution is shown alongside the adjusted distribution obtained via the difficulty-flip strategy. See Table 1 for a comparison of SFT results under different mixing strategies.

Table 1: Comparisons of SFT results with different difficulty-mixing strategies applied to the training dataset on 7B and 14B models. We report pass@1 accuracy of LiveCodeBench.

| Model | AIME24↑ | AIME25↑ | MATH500↑ | LiveCodeBench↑ |
|---|---|---|---|---|
| **Qwen2.5-Instruct-7B** | | | | |
| Bespoke-Stratos-17k | 10.0 | 6.7 | 77.2 | 33.8 |
| Difficulty-reverse-Flipped | 13.0 | 10.0 | 76.4 | 30.0 |
| Difficulty-Flipped (Ours) | **20.0** | **16.7** | **78.2** | **35.3** |
| **Qwen2.5-Instruct-14B** | | | | |
| Bespoke-Stratos-17k | 20.0 | 13.3 | 84.4 | 45.3 |
| Difficulty-reverse-Flipped | 20.0 | 23.3 | 83.0 | 43.8 |
| Difficulty-Flipped (Ours) | **23.0** | **23.3** | **84.4** | **45.5** |

To further investigate the role of difficulty in long CoT reasoning scenarios, we fine-tune Qwen2.5-7B-Instruct on the BS17k dataset and analyze influence scores grouped by difficulty levels. The results, shown in Figure 5(a), indicate that challenging tasks in both mathematics and coding are more beneficial for math reasoning. In contrast, easier math problems offer limited gains across both math and coding evaluations. This observation aligns with findings from the w/o long CoT setting and prior works such as LIMO [37], which highlight the utility of difficult math problems in developing reasoning capabilities. On the other hand, we find that simpler code problems are more effective for improving performance on coding tasks when mixed with math data. We hypothesize that, in addition to logical reasoning, programming tasks rely heavily on learning structural and syntactic patterns. When paired with math data that enhances logical thinking, simple coding tasks with clearer structure and more consistent syntax facilitate the model's acquisition of fundamental programming patterns, thereby improving code generation performance.

Based on these insights, we design an optimized data mixing strategy: we replace simple math problems in the original dataset with more challenging ones sourced from a larger scale OpenThoughts-114k[6] dataset, and conversely, we replace difficult coding problems with simpler ones. The modified dataset retaining the original size of 17k examples, compared in Figure 5(b), is used to retrain the model. As shown in Table 1, this new difficulty flipped mixing strategy yields consistent improvements across AIME, MATH, and LiveCodeBench benchmarks. In contrast, applying the reverse strategy—simplifying hard math problems and complicating easy coding tasks—results in the worst performance, further validating our finding.

---

[6]https://huggingface.co/datasets/open-thoughts/OpenThoughts-114k. Note that this dataset is curated using the same pipeline as BS-17k, with identical question sources and answers distilled from Deepseek-R1.

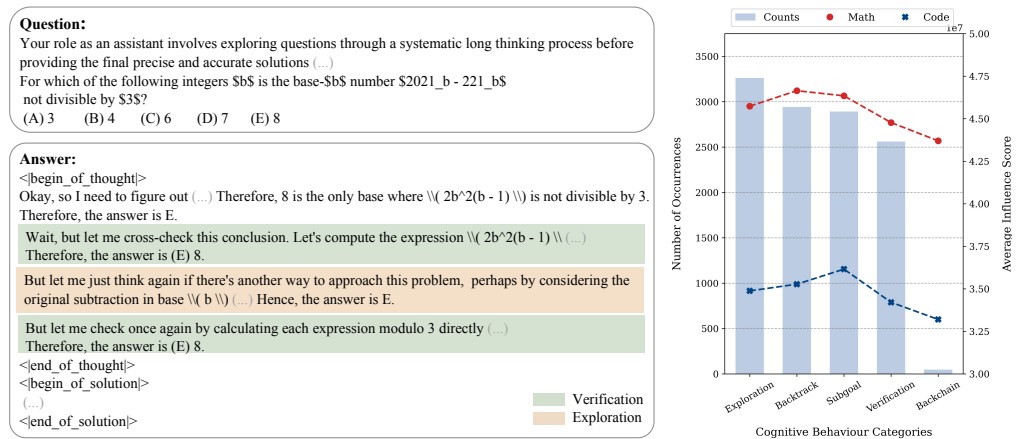

Figure 6: *Left:* An example of long CoT illustrating cognitive behaviors: verification (systematic error-checking) and exploration (searching for another approach after reaching the correct answer). *Right:* Distribution of different cognitive behaviors in BS-17k training dataset and their average impact on math and code reasoning performance.

Table 2: Sequence-level attribution of cognitive behaviors in long CoT. *Left:* Comparison of influence scores of the example in Figure 6 on math and code reasoning, w/ and w/o verification and exploration sentences. *Right:* Comparison of SFT results w/ and w/o exploration behaviors in BS-17k dataset.

| Domain | full CoT | w/o Ver. | w/o Exp. | w/o both | Model | MATH500 | LiveCodeBench |
|--------|----------|----------|----------|----------|--------|---------|---------------|
| Math | 2.2e+08 | 1.5e+08 | 9.0e+07 | 7.0e+07 | w/ Exp. | 77.2 | 33.8 |
| Code | 2.2e+07 | 1.6e+07 | 8.4e+06 | 7.5e+06 | w/o Exp. | 73.8 | 32.0 |

## 4.3 Sequence-level Attribution

> **Finding 3:**
>
> *The presence of 'searching for another approach after reaching correct answers' in math reasoning traces benefits to both math and code reasoning. While previously considered unnecessary overthinking, our sequence-level influence analysis and SFT ablations demonstrate its positive impact, suggesting such exploratory behaviour may promote generalizable reasoning skills.*

We are interested in the influence of different cognitive behaviors on reasoning performance. Following prior work [4], we focus on five key behaviors: exploration (seeking alternative approaches after reaching a correct answer), verification (systematic error-checking), backtracking (abandoning ineffective strategies), subgoal setting (breaking problems into manageable steps), and backward chaining (reasoning from desired outcomes to initial inputs). To identify these behaviors in the BS-17k dataset, we use Qwen-32B-instruct as a classifier, with details provided in Appendix E.

As shown in Figure 6 (right), exploration behavior is notably prevalent in the training dataset. However, prior research often views exploration as detrimental, considering it a form of overthinking that can reduce efficiency [2, 32]. We therefore seek to assess whether this cognitive behavior has a positive or negative impact using influence functions. Specifically, we truncate sentences in the training data where exploration behavior is present and examine the change in the influence score, as defined in Equation 6. The results, presented in Table 2 (left), show that exploration behavior is not redundant; on the contrary, it has a positive effect on both math and code reasoning performance, with the positive impact of exploration even exceeding that of verification.

To further validate this, we use GPT-4o to truncate all exploration behaviors in the BS-17k dataset for SFT, with the instruction details shown in Appendix D. The SFT results in Table 2 (right) show a significant performance drop when exploration behavior is removed. We attribute this decline to exploration's role in enabling flexible problem-solving, essential for adapting to diverse reasoning tasks. Beyond exploration, we compare average influence scores across other cognitive behaviors. As shown in Figure 6 (right), backtracking is crucial for mathematical reasoning, while subgoal setting

Figure 7: *Left:* Visualization of top 5% influential tokens in math CoT. *Right:* Visualization of top 5% influential tokens in code CoT.

is more impactful in programming tasks. This may be because programming requires breaking down high-level goals into modular components, making subgoal setting essential.

### 4.4 Token-level Attribution

> **Finding 4:**
>
> *Token-wise attribution analysis reveals distinct paradigms in math and code reasoning. In math CoT, influential tokens are natural language with logical connectors, whereas code CoT are dominated by structured code with syntax markers.*

To investigate the most influential tokens for stimulating reasoning, we select the top 100 highly influential examples on math and code reasoning, compute token-wise influence scores using Equation 8, and highlight the top 5% most influential tokens. Interestingly, as shown in Figure 7, the initial tokens in CoT—such as 'Okay, so I...'—are frequently highlighted, suggesting that these openers help orient the model's cognitive process to initiate reasoning. Further analysis reveals that, in math CoTs, the influential tokens are predominantly natural language logical connectors, such as 'Wait', 'However', 'Verify', 'Hence', 'First', 'Therefore', and 'Alternatively'. In contrast, in code CoTs, the most influential tokens are structural or syntactic elements such as markdown-style headings (e.g., ### Solution), fenced code blocks (e.g., ``` bash``` ), and syntax markers (e.g., def (self, A: List [int])-> int: ), which reflect the highly structured nature of code reasoning. This contrast highlights a divergence in reasoning paradigms: math reasoning relies more heavily on logical discourse, while code reasoning is facilitated by explicit structure and formatting. These divergent patterns may explain why easier code problems with clearer structural formats are particularly beneficial for enhancing code reasoning when integrated with math CoTs that already provide strong logical skills.

## 5   Conclusion

In this paper, we propose a fine-grained influence function framework to trace how training data on SFT phase shapes LLM reasoning in math and code tasks. Our analysis reveals that cross-domain examples—especially high-difficulty math and low-difficulty code—boost reasoning performance across domains. We further extend influence functions to the sequence level, revealing that exploratory behaviors in long CoT consistently enhance performance, challenging prior assumptions that such behaviors reflect overthinking. Token-level analysis reveals distinct paradigms in math and code reasoning. Our work highlights the utility of influence-based attribution for data-centric optimization and opens a path toward more targeted and interpretable reasoning supervised training.

**Limitations.** The main limitations of this work are as follows. We approximate the Hessian **H** by considering only the MLP parameters and treating the attention as fixed to approximate influence functions for simplicity. Besides, our analysis is limited to mathematical and coding reasoning tasks; extending this framework to other domains, such as commonsense reasoning, remains an open direction for future work.

## Acknowledgements

This work was supported by NSF of China (Nos. 92470118, 62306176), Natural Science Foundation of Shanghai (No. 23ZR1428700), CCF-ALIMAMA TECH Kangaroo Fund (NO. CCF-ALIMAMA OF 2025010), and CCF-Zhipu Large Model Innovation Fund (No. CCF-Zhipu202412).

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

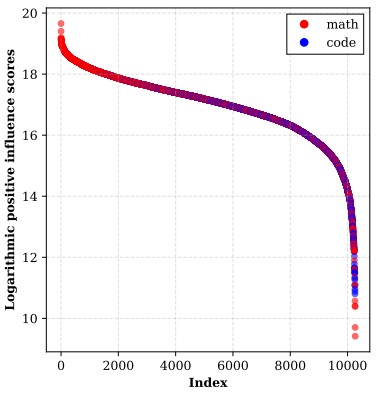
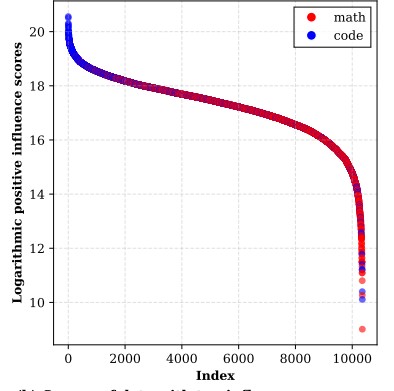

(a) Source of data with top influence scores on math     (b) Source of data with top influence scores on code

Figure 8: Cross-domain influence analysis of Qwen2.5-7B-Instruct fine-tuned on Bespoke-Stratos-17k dataset for math and code reasoning performance.

## A Derivation of Influence Score

Given the influence of $z_m$ on model parameters $\boldsymbol{\theta}$

$$\mathcal{I}_{\boldsymbol{\theta}}(z_m) = \left.\frac{d\boldsymbol{\theta}}{d\epsilon}\right|_{\epsilon=0} = -\mathbf{H}^{-1}\nabla_{\boldsymbol{\theta}}\mathcal{L}(z_m, \boldsymbol{\theta}),\tag{9}$$

we can obtain its influence on a function of parameters $f(\boldsymbol{\theta})$ by applying the chain rule for derivatives:

$$\begin{aligned}\mathcal{I}_f(z_m) &= \left.\frac{df(\boldsymbol{\theta})}{d\epsilon}\right|_{\epsilon=0}\\ &= \nabla_{\boldsymbol{\theta}}f(\boldsymbol{\theta})^T \left.\frac{d\boldsymbol{\theta}}{d\epsilon}\right|_{\epsilon=0}\\ &= -\nabla_{\boldsymbol{\theta}}f(\boldsymbol{\theta})^T\mathbf{H}^{-1}\nabla_{\boldsymbol{\theta}}\mathcal{L}(z_m, \boldsymbol{\theta}).\end{aligned}\tag{10}$$

## B Cross Domain Influence Analysis in Long CoT Scenarios

In this section, we provide additional instance-level attribution experiment on long CoT reasoning scenarios. We fine-tune Qwen2.5-7B-Instruct on Bespoke-Stratos-17K reasoning dataset. As shown in Figure 8(a), the most influential samples for improving math performance predominantly from the math domain, but the samples from code domain are also significant. In Figure 8(b), there is a similar pattern of cross-domain benefit. This is consistent with the conclusions we obtained in the experimental section 4.2.

## C Robustness on $n$

In this section, we evaluate the robustness of the influence function estimates with respect to the size of the correct subset $\mathcal{D}_{\text{correct}}$. Specifically, we fine-tune the LLaMA3-8B-Base model on a mixed training corpus comprising MetaMathQA and OSS-Instruct, and compute influence scores on the math and code performance. We calculate the Pearson correlation between the rankings of training examples induced by influence scores using varying values of $n$, using $n = 100$ as the reference. Results in Table 3 shows the robustness of $n$ for influence scores estimation.

Table 3: Pearson correlation coefficient of rankings on training data across different choices of $n$, indicating stable influence estimation.

| $n \rightarrow$ | **10** | **25** | **50** | **100** |
|---|---|---|---|---|
| Math | 0.52 | 0.60 | 0.70 | 1.0 |
| Code | 0.51 | 0.62 | 0.60 | 1.0 |

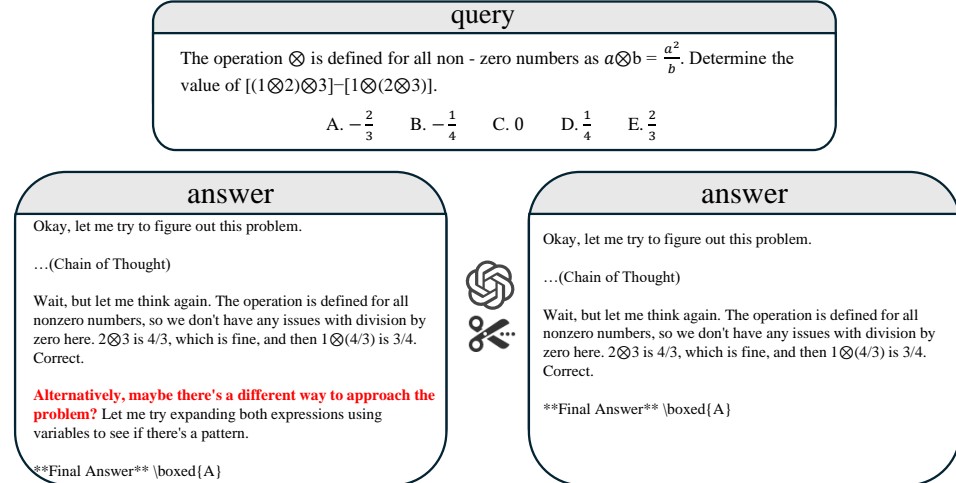

Figure 9: To assess whether the exploration behavior has a positive or negative impact, we use GPT-4o to truncate all exploration behaviors in the BS-17K dataset for SFT. If reasoning contains any searching for another approach after reaching correct answers, like "Alternatively, maybe there's a different way to approach the problem?", the exploration content will be truncated.

## D Case of Truncating Exploration Behavior

To evaluate the impact of exploration behaviors in reasoning processes, we systematically truncate exploratory content from the BS-17K during SFT. Specifically, any post-correct-answer exploration (e.g., "Alternatively, maybe there's a different way to approach the problem?") is removed to isolate the core problem-solving trajectory, as shown in Figure 9.

## E Examples for Reasoning Behaviors Classifier

The five cases below show the prompts of five behaviors on reasoning performance and the corresponding answers. As shown in Figure 10 and 11, the prompts include task description, examples of each reasoning behavior, task format, etc.

| Exploration | Verification |
|---|---|
| ***Prompt:***

#Task Description
You will be provided with text from the internet.
Evaluate whether the text contains any "searching for another approach after reaching correct answers" behavior, where the writer reaches a correct or satisfactory solution, but then explicitly continues to explore alternate methods or perspectives. This is a type of cognitive behavior that reflects intellectual curiosity or seeking multiple strategies even after solving the problem.

This is not the same as:
- Backtracking, where the writer changes method because the current one does not work or is incomplete.
- Verification,where the writer checks the correctness of a method without attempting a different one.

Examples of this behavior include:
- "But let me just think again if there's another way to approach this."
- "Another way to look at this is..."
- "That's one way to get the result. Now let's see if we can also derive it using geometry."
- "Alternatively, maybe there's a different way to approach the problem?"
Count the number of distinct backtracking instances and provide the count between the tags <count> </count>. If the writer does not exploration, please provide a count of 0 as <count>0</count>.

#Task Format
Format your response in markdown as follows:

##Thoughts
[Brief description describing what behavior was noticed and where exploration occurred]

##Does exploration thinking?
[yes/no]

##Number of exploration thinking steps
<count> [1/2/...] </count>

#Task to evaluate for exploration
{response}

#Response | ***Prompt:***

# Task Description
You will be provided with text from the internet.
Evaluate whether the text contains any verification steps. We want to mark instances where the writer explicitly checks their own work, such as by comparing the result to a known value or by checking the result of a calculation.

Verification steps in mathematics might look like:
- "Let's check our answer by substituting x = 3 back into the original equation."
- "To verify this is correct, I'll differentiate the antiderivative and confirm it matches the original function."
- "Let's test our formula with a simple case: when n = 1, we get f(1) = 2, which matches our expected result."
- "To ensure this solution is valid, I'll check if it satisfies all the given constraints."

If you find any verification steps, please count them and provide the count between the tags <count> </count>. If the text does not contain any verification steps, please provide a count of 0 as <count>0</count>.

# Task Format
Format your response in markdown as follows:

## Thoughts
[Brief description describing what behavior was noticed and where answer verification may have occurred]

## Does verification?
[yes/no]

## Number of answer verification steps
<count> [1/2/...] </count>

# Task to evaluate for Verification
{response}

# Response |
| ***Answer:***
##
Thoughts
The writer explores multiple ways to approach the problem of proving that a sequence satisfying certain conditions must be a polynomial. After initially considering the properties of polynomials and the given conditions, the writer continues to explore different methods and perspectives, such as using finite differences, induction, and theorems about integer-valued polynomials. The writer does not stop at the first satisfactory approach but continues to question and explore further, indicating a behavior of seeking alternative methods even after reaching a correct or satisfactory solution.

##
Does alternative thinking?
yes

##
Number of alternative thinking steps

<count> 5 </count> | ***Answer:***
##
Thoughts
The provided text is a detailed exploration of a mathematical problem involving sequences and polynomials. The writer discusses various approaches and theorems that could be used to prove that a sequence satisfying certain conditions must be a polynomial. However, the text does not explicitly check any calculations or results against known values or re-verify any steps taken in the reasoning process.

##
Does verification?
no

##
Number of verification steps

<count> 0 </count> |

Figure 10: *Left:* Exploration: When performing reasoning, seeking alternative approaches after reaching a correct answer. We capture this behavior and calculate the number of exploration steps by analyzing the content like "Another way to look at this is..." etc.. *Right:* Verification: The behavior of reasoning from desired outcomes to initial inputs when performing reasoning. We capture and calculate the number of backward chaining instances by finding the content like "To solve this equation, let's start with what we want to prove" etc..

| Backtracking | Backward Chaining |
|---|---|
| **Prompt:**

# Task Description
You will be provided with text from the internet.
Evaluate whether the text contains any backtracking behavior, where the writer realizes a path won't work and explicitly goes back to try a different approach. An example of backtracking is: "Let me try again", "Wait", "I made a mistake", or "we need to try a different sequence of operations". We want to mark instances where the writer abandons a thought and backtracks to a previous computation.

Backtracking in mathematics might look like:
- "I started with the wrong formula. Let's use integration by parts instead."
- "This approach leads to a contradiction. Going back to the original equation..."
- "I see the error in my calculation. Let's recalculate using..."
- "This algebraic manipulation isn't simplifying as expected. Let's try factoring differently."

Count the number of distinct backtracking instances and provide the count between the tags <count> </count>. If the writer does not backtrack, please provide a count of 0 as <count>0</count>.

# Task Format
Format your response in markdown as follows:

## Thoughts
[Brief description describing what behavior was noticed and where backtracking occurred]

## Does backtrack?
[yes/no]

## Number of backtrack steps
<count> [1/2/...] </count>

# Task to evaluate for backtracking
{response}

# Response | **Prompt:**

# Task Description
You will be provided with text from the internet.
Evaluate whether the text contains any backward-chaining behavior, where the writer is working towards a goal but starts from the goal and works backward.

Backward-chaining in mathematics might look like:
- "To solve this equation, let's start with what we want to prove: x = 4. Working backward, if x = 4, then x² - 5x + 4 = 0 must be true. Let's verify this."
- "If we want to find a function whose derivative is 2x + 3, let's start with the desired result and work backward.
The derivative of x² + 3x + C would give us 2x + 3."
- "To prove this triangle is a right triangle, I'll start by assuming the Pythagorean theorem holds (c² = a² + b²) and check if our side lengths satisfy this condition."
- "In this optimization problem, we know the maximum value occurs when the derivative equals zero. If we set f'(x) = 0 and solve for x, we can find our critical points."
- "To determine the initial velocity, I know the ball reaches a height of 20 meters. Working backward from this final state using the kinematic equation h = vt - 0.5gt², I can solve for the initial velocity v."

Count the number of distinct backward-chaining instances and provide the count between the tags `<count> </count>`. If the reasoning does not contain any backward-chaining behavior, provide a count of 0 as `<count>0</count>`.

# Task Format
Format your response in markdown as follows:

## Thoughts
[Brief description of the observed behavior and where backward chaining may have occurred]

## Does the text exhibit backward chaining?
[yes/no]

## Number of backward chaining instances
<count> [1/2/...] </count>

# Task to evaluate for backward chaining
{response}

# Response |
| **Answer:**
##
Thoughts
The writer initially explores the problem by considering the conditions given and trying to connect them to known properties of polynomials. They then realize that their initial approach might not be leading to a solution and start to consider alternative methods, such as using the method of finite differences, induction, and other properties of polynomials. The writer explicitly mentions reconsidering their approach multiple times, indicating backtracking behavior.

##
Does backtrack?
yes

##
Number of backtrack steps

<count> 4 </count> | **Answer:**
##
Thoughts
The provided text does not exhibit backward-chaining behavior. The writer is attempting to prove a statement about a sequence of integers satisfying certain conditions, but the reasoning is forward-looking, starting from the given conditions and working towards a conclusion. The writer explores various mathematical concepts and theorems to understand the problem and does not start from a goal and work backward.

##
Does the text exhibit backward chaining?
no

##
Number of backward chaining instances

<count> 0 </count> |

Figure 11: *Left:* Backtracking: The behavior of realizing a path won't work and explicitly going back to try a different approach. We capture this behavior and calculate the number of backtracking steps by finding the content like "This approach leads to a contradiction. Going back to the original equation..." etc.. *Right:* Backward Chaining: The behavior of systematic error-checking when performing reasoning. We capture and calculate the number of backward chaining instances by finding the content like "To ensure this solution is valid, I'll check if it satisfies all the given constraints." etc..

