# OpenReview forum: "Which Data Attributes Stimulate Math and Code Reasoning? An Investigation via Influence Functions"
_NeurIPS.cc/2025/Conference — NeurIPS 2025 poster_

### Official Review · Reviewer_PJJ6 · 2025-06-10

**Clarity:** 3
**Significance:** 2
**Originality:** 2
**Rating:** 4
**Confidence:** 4

**Summary:**

This paper uses influence functions (EK-FAC) to quantify the influence between math and code training examples on math and code tasks. They find substantial cross-domain influence, where high-difficulty math examples improve both math and code reasoning, while low-difficulty code examples are beneficial for code reasoning. At the token level, logical connectors tend to be the most influential for math reasoning, while syntactic tokens tend to be most influential for code reasoning. They use these insights to tune Qwen models on more high-difficulty math examples (with low-difficulty code examples) to improve math and code reasoning performance.

**Questions:**

1. Is there a way to distinguish whether the observed influence scores are primarily driven by surface level similarities between examples (e.g. difficult math problems may tend to include similar words or phrases to one another and to coding examples), vs. more abstract "procedural" similarities (e.g. https://arxiv.org/abs/2411.12580)?
2. Could the tuning results on the proposed "Difficulty Flipped" dataset be because that data has substantially more "difficult" problems overall than all the other comparison datasets (see Weaknesses)?
3. Is there evidence that the high-difficulty math examples are highly influential because they contain important reasoning strategies that the model relies on for other math and code reasoning problems? Are there ways to characterize or classify these different reasoning strategies? More discussion and results on this point could make the paper more impactful (and improve my scores), because it would help clarify exactly why these types of examples are so important.

**Ethical Concerns:**

["NO or VERY MINOR ethics concerns only"]

**Final Justification:**

The author response has addressed several of my questions, and I have increased my overall score to 4.

**Limitations:**

Yes

**Quality:**

2

**Strengths And Weaknesses:**

Strengths:
1. Quantifying the relationship between math and code examples using influence functions has direct downstream implications for LLM fine-tuning, as performed in the paper.

Weaknesses:
1. The new method ("Infra") might have somewhat limited novelty. It seems very close to an implementation of previous influence approaches (specifically, EK-FAC; https://arxiv.org/abs/2308.03296, as cited in the paper). The main differences are slightly changing the response function f (although f is still essentially log-probabilities), and applying to new input/output examples.
2. The tuning results on the proposed "Difficulty Flipped" dataset are not entirely convincing, because that data has substantially more "difficult" problems overall than all the other comparison datasets (Figure 5, right). It is somewhat expected that including more difficult problems in the last tuning phase would benefit performance.
3. Other works have applied influence functions to LLMs for reasoning and/or mathematical tasks (e.g. https://arxiv.org/abs/2411.12580, https://arxiv.org/abs/2410.17413). These works have demonstrated that examples with similar semantic patterns (e.g. similar reasoning) are often influential, so cross-domain influence between math and code problems is not entirely surprising. More fine-grained breakdowns (e.g. annotating examples that exhibit similar reasoning strategies) could make the paper more informative, by isolating exactly why specific math and code examples influence one another.

---

> ### Author Rebuttal · Authors · 2025-07-31
>
> Thank you for your attentive comments and useful suggestions! We are glad you thought our paper had direct downstream implications for LLM fine-tuning. We address your concerns point by point below.
>
> **W1:** The new method ("Infra") might have somewhat limited novelty. It seems very close to an implementation of previous influence approaches (specifically, EK-FAC; https://arxiv.org/abs/2308.03296, as cited in the paper).
>
> Thank you for your feedback. We would like to emphasize that **our main contribution lies in the novel application of influence functions to trace how training data during the SFT phase shapes LLM reasoning for the first time**, presenting key findings with substantial practical value, as acknowledged by Reviewer mqmh.  Reviewer AwiZ also noted that we 'target an important problem and uncover multiple interesting findings', and Reviewer x5z4 praised our work for 'the analysis fills a critical gap that needed to be addressed'.
>
> Furthermore, unlike prior approaches, our work introduces **a fine-grained analysis**: sequence-level analysis reveals the counterintuitive insight that "exploration" behavior can enhance reasoning, while token-level analysis provides valuable insights into the interaction mechanisms between math and code training data.
>
> **W2 & Q2:**  It is somewhat expected that including more difficult problems in the last tuning phase would benefit performance. Could the tuning results on the proposed Difficulty Flipped dataset be because that data has substantially more difficult problems overall than all the other comparison datasets?
>
> Indeed, one of our key findings is that high-difficulty math examples improve both math and code reasoning, and therefore, the performance improvements observed in the Difficulty Flipped dataset can partly be attributed to the inclusion of more difficult math problems.
>
> The reviewer mentioned the intuition that difficult problems may lead to better performance. This can be wrong in practice, e.g., for the code capacities. To validate this, **we conduct an ablation study on Qwen2.5-Instruct-7B, with both high-difficulty math and code samples for training. The results in the table below show a decline in coding performance compared to the proposed Difficulty Flipped dataset.** This may be due to the fact that easier code problems, with their clearer structural formats, are more effective for enhancing code reasoning, especially when integrated with math CoTs that already provide strong reasoning skills.
>
> Thank you for pointing this out, and we will add this ablation result to the revised version.
>
> |                    | AIME24$\uparrow$ | MATH500$\uparrow$ | LiveCodeBench$\uparrow$ |
> |--|--|--|--|
> | BS-17k                     | 10.0   | 77.2    | 33.8           |
> | Difficulty-Flipped (Hard Math + Easy Code) | 20.0 | 78.2 | 35.3 |
> | Hard Math + Hard Code      | 20.0   | 78.0    | 34.5           |
>
> **W3:** Other works have applied influence functions to LLMs for reasoning and/or mathematical tasks (e.g. https://arxiv.org/abs/2411.12580, https://arxiv.org/abs/2410.17413). More fine-grained breakdowns could make the paper more informative, by isolating exactly why specific math and code examples influence one another.
>
> Thanks for pointing out this valuable direction for deeper interpretability of cross-domain influence. First, we would like to clarify that these works primarily focus on attributing LLM performance to pretraining data, whereas our work targets the SFT phase. We will include citations to these works in the revised version.
>
> Regarding more fine-grained breakdowns, as shown in our paper (Figure 3(a) and Figure 4), we find that **symbolic math problem-answer pairs introducing or solving for x are among the most beneficial for code-related tasks**. This may be because the process of reasoning with symbolic variables in math closely mirrors the patterns used in programming tasks.
>
> To further explore long-CoT reasoning settings, we conduct a case study analyzing the top 20 math examples with the highest impact on LiveCodeBench performance. Our observations suggest that **the most influential math problems are predominantly symbolic manipulation tasks requiring equation rearrangement and factorization** (see some examples below), likely because they cultivate algebraic reasoning and variable isolation skills that are fundamental for implementing mathematical algorithms and debugging complex code expressions. We will include this in the revised version.
>
> ```
> Question: If $b>1$, $x>0$, and $(2x)^{\\log_b 2}-(3x)^{\\log_b 3}=0$, then what is $x$?
> Answer(partial): So, if I take \\( \\log_b \\) of both sides:
> \\( \\log_b \\left( (2x)^{\\log_b 2} \\right) = \\log_b \\left( (3x)^{\\log_b 3} \\right) \\).
> Using the logarithm power rule, \\( \\log_b (a^c) = c \\cdot \\log_b a \\), so applying that here, the equation becomes:
> \\( \\log_b 2 \\cdot \\log_b (2x) = \\log_b 3 \\cdot \\log_b (3x)
> ```
> ```
> Question: If $x, y, z$ are reals such that $0\\le x, y, z \\le 1$, show that $\\frac{x}{y + z + 1} + \\frac{y}{z + x + 1} + \\frac{z}{x + y +  1} \\le 1 - (1 - x)(1 - y)(1 - z)$
> Answer(partial): Let's denote S = x + y + z, P = xy + yz + zx, Q = xyz. Then RHS is S - P + Q.
> The LHS is sum of x\/(S - x + 1). Wait, because denominator for the term with x is y + z + 1 = (y + z) + 1 = (S - x) + 1. So LHS can be written as: Sum_{x} [x \/ (S - x + 1)].
> ```
>
> **Q1:** Is there a way to distinguish whether the observed influence scores are primarily driven by surface-level similarities between examples, vs. more abstract "procedural" similarities (e.g. https://arxiv.org/abs/2411.12580)?
>
> Thank you for raising this insightful question. To assess whether influence scores are primarily driven by surface-level similarities, we compute BERTScore[1] between each of the 100 AIME and MATH500 examples and their top-10 most influential training examples. BERTScore computes semantic similarity between tokens in a candidate sentence and a reference sentence using contextual embeddings, making it suitable for assessing surface-level overlap.
>
> We report the maximum BERTScore(F1) among the top-10 influential examples for each target question (across 100 total targets) in the table below. Most maximum scores are around 0.65, suggesting that **the influence scores are unlikely to be dominated by superficial lexical overlap**, but more likely reflect abstract procedural patterns--consistent with findings in https://arxiv.org/abs/2411.12580.
>
> [1] BERTScore: Evaluating Text Generation with BERT, ICLR 2020
> |  | 1    | 2    | 3    | 4    | 5    | 6    | 7    | 8    | 9    | 10   |
> |-|-|-|-|-|-|-|-|-|-|-|
> |  max      | 0.62 | 0.65 | 0.58 | 0.7  | 0.68 | 0.6  | 0.55 | 0.61 | 0.68 | 0.63 |
> |  mean |  0.42 | 0.45 | 0.37 | 0.41 | 0.41 | 0.39 | 0.36 | 0.40 | 0.46 | 0.39 |
>
> **Q3:** Is there evidence that the high-difficulty math examples are highly influential because they contain important reasoning strategies that the model relies on for other math and code reasoning problems? Are there ways to characterize or classify these different reasoning strategies? More discussion and results on this point would help clarify exactly why these types of examples are so important.
>
> Thank you for this interesting question for the deeper interpretation of why high-difficulty math examples are highly influential. To investigate this, we compare 100 high-difficulty examples with top influence scores and 100 low-difficulty examples with top influence scores. We observe that high‑difficulty samples consistently exhibit (1) **Knowledge Integration**, where solutions draw on concepts from multiple domains, and (2) **Strategy Diversity**, where diverse reasoning patterns are employed within a single solution. We use GPT-4o based annotation (prompt templates are shown below) to quantify these phenomena. As shown in the table below, both knowledge integration and strategy heterogeneity occur significantly more often in high‑difficulty examples, which likely underlies their better transferability and generalization to other reasoning problems.
>
> |     | High difficulty | Low difficulty |
> | - | -| - |
> | Knowledge Integration | 0.78 | 0.34 |
> | Strategy Diversity | 0.91 | 0.76 |
>
> ```
> **Task Description**
> You will be given a piece of text. Please evaluate whether this text exhibits knowledge integration. Knowledge integration occurs when multiple domains or disciplines of knowledge and strategies are combined to solve a problem effectively—for example, blending algebra, geometry, and probability in a single math solution. Look specifically for instances where the text weaves together concepts from different fields.
> **Examples**
> “To solve the variance problem, we first wrote down the expectation and second-moment formulas using probability theory, then applied algebraic manipulation to solve the resulting quadratic equation for p.”
> **Output Format**
> ## Thoughts
> [Briefly note what you observed that indicates knowledge integration, and which disciplines or techniques are being combined.]
>
> ## Does knowledge integration occur?
> [yes / no]
> ```
>
> ```
> **Task Description**
> You will be given a piece of text. Please evaluate whether this text exhibits strategy diversity. Strategy diversity means using three or more distinct problem-solving methods or reasoning paths—such as decomposition, recursion, dynamic programming, greedy selection, etc.—within the same solution. Focus on spotting multiple, clearly different strategies being applied in tandem or in sequence.
> **Examples**
> “First we apply a divide-and-conquer approach to break the problem into subproblems, then we optimize each subproblem via dynamic programming.”
> **Output Format**
> ## Thoughts
> [Briefly note what you observed that indicates strategy diversity, and which disciplines or techniques are being combined.]
>
> ## Does strategy diversity occur?
> [yes / no]
> ```
>
> Given these improvements and additional experiments to your initial concerns, we hope you would like to raise your score.

---

> > ### Comment · Reviewer_PJJ6 · 2025-08-01
> >
> > Thank you for the responses! These address several of my questions, and I have increased my overall score to 4.

---

> > > ### Author Response · Authors · 2025-08-07
> > >
> > > Thank you again for your valuable feedback and for the improved score! We will incorporate your suggestions in the final version.

---

### Official Review · Reviewer_mqmh · 2025-06-28

**Clarity:** 3
**Significance:** 3
**Originality:** 3
**Rating:** 5
**Confidence:** 4

**Summary:**

This paper delves into what types of data are beneficial for enhancing mathematical and code reasoning capabilities during the SFT (Supervised Fine-Tuning) stage. Specifically, the authors propose three levels of influence scores and utilize influence functions to investigate the relationship between training data and downstream performance. They arrive at four meaningful conclusions. Overall, the work is logically coherent, and the conclusions are persuasive and potentially of practical value.

**Questions:**

1. Do the authors offer any preliminary exploration or conclusions regarding the internal mechanisms of large language models (LLMs)?

2. The conclusion of the paper seems somewhat rushed. Are there any additional key experiments or findings that should have been included?

3. How was the design of the influence scores determined? What was the rationale behind this choice?

4. Does replacing the Hessian matrix with the EK-FAC method compromise the reliability of the conclusions? Have the authors compared the impact of other alternatives on the final results?

**Ethical Concerns:**

["NO or VERY MINOR ethics concerns only"]

**Final Justification:**

I will maintain my initial positive rating because the work presents interesting conclusions and solid experiments, and the authors also provided valid rebuttals during the rebuttal.

**Quality:**

3

**Strengths And Weaknesses:**

Strengths:

1. This work explores and explains the underlying mechanisms behind the success of language models and presents key findings with substantial practical value and guiding potential.

2. The relatively comprehensive experiments provide strong support for the reliability of the conclusions.

3. The paper follows a clear line of thought, has a well-structured logic, and is written in a way that is easy to understand.

Weakness:

1. The conclusions of this work are confined to the SFT stage, whereas it is widely acknowledged that the capabilities of large language models primarily stem from the pre-training phase. Merely exploring the sources of capability during the SFT stage may be far from sufficient. That said, I understand this limitation, as investigating the origins of model capabilities during pre-training is extremely complex and resource-intensive.

2. The design of the influence scores appears to be overly intuitive, lacking the necessary theoretical or empirical justification to support the reasonableness of such a formulation.

3. This paper examines the impact of data on model capabilities solely from the perspective of training data → performance, resulting in conclusions that remain relatively superficial. It lacks deeper investigation and understanding of the internal mechanisms of the model.

---

> ### Author Rebuttal · Authors · 2025-07-31
>
> Thank you for the positive feedback and useful suggestions! We address your concerns point by point below.
>
> **W1:** The conclusions of this work are limited to the SFT stage, while model capabilities largely stem from pre-training. Although this focus may be insufficient, the complexity and cost of analyzing pre-training make it understandable.
>
> Thank you for your feedback. Indeed, it is widely acknowledged that the capabilities of LLMs are learned during pre‑training. However, it is widely recognized that the SFT phase is indispensable for unlocking and shaping those capabilities [1,2]. Thus, characterizing which properties of SFT data most effectively stimulate reasoning remains highly valuable. We acknowledge that  investigating the pre-training phase would be illuminating, but it is beyond the scope of this work due to its complexity and scale; we therefore propose such an investigation as important future work.
>
> [1] Towards Reasoning in Large Language Models: A Survey, ACL 2023
>
> [2] Octothinker: Mid-training incentivizes reinforcement learning scaling. arXiv:2506.20512
>
> **W2 & Q3:** The design of the influence scores lacks the necessary theoretical or empirical justification to support the reasonableness of such a formulation. What was the rationale behind the design of the influence scores?
>
> The influence score formulation is grounded in well-established theoretical foundations. It originates from the work [1], which rigorously derives how the removal of a training example affects model parameters and, in turn, model predictions. This approach provides a principled framework for quantifying the impact of individual training samples on model behavior.
>
> In our work, we design the influence scores by adopting the mean log-probabilities on the correct answers as the response function f, because this metric closely reflects the model's reasoning ability. We appreciate this question and will include these clarifications in the revised version.
>
> [1] Understanding Black-box Predictions via Influence Functions, ICML 2017
>
> **W3 & Q1:** This paper examines the impact of data on model capabilities solely from the perspective of training data → performance, lacking deeper investigation and understanding of the internal mechanisms of the model.
>
> Our primary focus is indeed on understanding how training data influences model performance, which we believe is both scientifically meaningful and practically valuable—especially for guiding data curation and mixture strategies. Reviewer AwiZ also noted that we 'target an important problem and uncover multiple interesting findings'.
>
> While we agree that probing the internal mechanisms of LLMs is important, it remains a highly challenging task due to the complexity and scale of these models. Nevertheless, our influence-based approach provides a potential pathway for such analysis. For instance, by decomposing the gradients involved in the influence computation, one could potentially attribute specific layers or modules within the model to reasoning performance. We consider this an interesting direction and leave it as future work.
>
> **Q2:** Are there any additional key experiments or findings that should have been included in the conclusion?
>
> We would like to clarify that the current conclusion section has systematically summarized our key findings:
>   - Cross-domain examples—particularly high-difficulty math and low-difficulty code—enhance reasoning performance across domains.
>   - Sequence-level analysis reveals that exploratory behaviors in long CoT consistently improve performance, challenging assumptions that such behaviors are merely overthinking.
>   - Token-level analysis further uncovers domain-specific reasoning patterns between math and code.
>
> **Q4:** Does replacing the Hessian matrix with the EK-FAC method compromise the reliability of the conclusions? Have the authors compared the impact of other alternatives on the final results?
>
> Thanks for pointing this out. Computing the exact Hessian is intractable for LLMs due to the prohibitive memory and computational costs. While K-FAC is a commonly used approximation, prior work has shown that it is less accurate than EK-FAC [1]. Moreover, EK-FAC has been shown to offer a good trade-off between computational efficiency and accuracy [1,2], which supports the reliability of our conclusions. Additionally, our SFT experiments (Table 1 & Table 2b) further validate the robustness of these findings.
>
> [1] Fast Approximate Natural Gradient Descent in a Kronecker-factored Eigenbasis, NIPS 2018
>
> [2] Studying Large Language Model Generalization with Influence Functions, arXiv:2308.03296

---

> > ### Comment · Reviewer_mqmh · 2025-08-09
> >
> > I appreciate the author's responses, most of which make sense. This has strengthened my confidence in the work, so I will maintain a positive score.

---

> > > ### Author Response · Authors · 2025-08-09
> > >
> > > Thank you for your feedback!

---

### Official Review · Reviewer_AwiZ · 2025-07-01

**Clarity:** 3
**Significance:** 2
**Originality:** 2
**Rating:** 4
**Confidence:** 4

**Summary:**

This paper proposes the Infra framework, which uses influence functions to quantify the impact of training data on the mathematical and code reasoning capabilities of large language models (LLMs). The core findings include: high-difficulty mathematical data significantly improves code reasoning, and low-difficulty code data is most beneficial for code tasks; ’ searching for another approach after reaching correct answers’ in math reasoning traces benefits to both math and code reasoning; and Token-wise attribution analysis reveals distinct paradigms in math and code reasoning.

**Questions:**

First, my major concern is that using the influence function to assign weights to data samples is similar to data selection. Thus, there lacks the comparison (experimentally or just discussion) between this solution with different data selection methods [1, 2].

[1] LESS: Selecting Influential Data for Targeted Instruction Tuning. ICML 24
[2] SmallToLarge (S2L): Scalable Data Selection for Fine-tuning Large Language Models by Summarizing Training Trajectories of Small Models. NeurIPS 24

Then, the influence function mainly relies on EK-FAC, which is not novel. The limitation of EK-FAC needs to be discussed.

Even though multiple interesting findings have been reported, they come from one-time experiments run. As a study paper, the authors need to run the experiments multiple times and then conduct a statistical analysis for a rigorous results report.

**Ethical Concerns:**

["NO or VERY MINOR ethics concerns only"]

**Final Justification:**

The response addressed most of my reviews.

**Limitations:**

yes

**Quality:**

3

**Strengths And Weaknesses:**

Strength:

+ Target an important problem – analyzing how data samples affect the performance of LLMs from different levels, e.g., sequence level and token-level.
+ Multiple interesting findings
+ A new data reweighting strategy that can help build better fine-tuning datasets

Weakness:

- Lack of comparison with data selection methods
- The proposed solution highly relies on EK-FAC, but lacks a limitation discussion of EK-FAC
- The experimental results are not reported rigorously, and there is no statistical analysis.

---

> ### Author Rebuttal · Authors · 2025-07-31
>
> Thank you for the feedback and useful suggestions! We are glad you thought our paper 'targeted an important problem and had multiple interesting findings'. We address your concerns point by point below.
>
> **W1 & Q1:** Lack of comparison (experimentally or just discussion) with data selection methods.
>
> Thank you for your feedback. We would like to emphasize that our work is not for data selection, which seeks a minimal subset of training examples for efficient domain transfer. Rather, our primary goal is different: we employ influence functions to attribute model performance back to individual training examples, thereby revealing which data types drive cross-domain reasoning improvements, which can in turn inform the design of more effective data curation and mixture strategies.
>
> Moreover, **influence-based sequence/token-level analysis enables us to ask why certain samples or patterns are helpful—e.g., identifying specific reasoning patterns that correlate with high influence—which is difficult to achieve with data selection methods** like LESS[1] and S2L[2]. We argue that this analytical perspective offers complementary insights that could help guide future improvements in training data design.
>
> [1] LESS: Selecting Influential Data for Targeted Instruction Tuning. ICML 24
>
> [2] SmallToLarge (S2L): Scalable Data Selection for Fine-tuning Large Language Models by Summarizing Training Trajectories of Small Models. NeurIPS 24
>
> **W2 & Q2:** The influence function mainly relies on EK-FAC, which is not novel. The limitation of EK-FAC needs to be discussed.
>
> Thanks for pointing this out. Computing the exact Hessian is intractable for LLMs due to prohibitive memory and computational requirements. Among available approximations, EK-FAC [1] has been shown to provide a good trade-off between computational efficiency and accuracy. Nevertheless, we acknowledge its limitations and will add this discussion to the revised version.
>   - **Block-Diagonal and Kronecker Factorization Assumptions.** It assumes that different layers are independent, ignoring inter-layer curvature interactions. Moreover, it assumes each layer’s Fisher block is exactly separable as a Kronecker product, which may poorly capture complex curvature.
>   - **Increased computation overhead than K-FAC.** EK‑FAC incurs approximately twice the computational cost compared to K‑FAC by forming and inverting two Kronecker factors and performing their eigendecompositions.
>
> [1] Fast Approximate Natural Gradient Descent in a Kronecker-factored Eigenbasis, NIPS 2018
>
> **W3 & Q3:** As a study paper, the authors need to run the experiments multiple times and then conduct a statistical analysis for a rigorous results report.
>
> We would like to clarify that **the influence‑score computation (Equation 3) is entirely deterministic, and thus there is no randomness in the results related to influence scores**. For SFT results in Tables 1 and 2(b), we retrain models with two additional random seeds and report the resulting standard deviations (shown in parentheses) for robustness. These new results will be included in the revised version. Thank you for your suggestion.
> | Qwen2.5-Instruct-7B       | AIME24$\uparrow$     | AIME25$\uparrow$     | MATH500$\uparrow$    | LiveCodeBench$\uparrow$ |
> |---------------------------|------------|------------|------------|----------------|
> | Bespoke–Stratos–17k       | 10.0 (1.1) | 6.7 (0.9)  | 77.2 (0.2) | 33.8 (0.3)     |
> | Difficulty–Flipped        | 20.0 (1.8) | 16.7 (1.1) | 78.2 (0.1) | 35.3 (0.1)     |
>
> |               | MATH500$\uparrow$    | LiveCodeBench$\uparrow$ |
> |---------------|------------|----------------|
> | w/ Exp.       | 77.2 (0.2) | 33.8 (0.3)     |
> | w/o Exp.      | 73.8 (0.5) | 32.0 (0.1)     |
>
> Given these clarifications and improvements to your initial concerns, we hope for your reconsideration in raising your score.

---

> > ### Comment · Reviewer_AwiZ · 2025-08-05
> >
> > The response addressed most of my concerns, I increased my score.

---

> > > ### Author Response · Authors · 2025-08-07
> > >
> > > Thank you for considering our response and for increasing the score. We are grateful for your feedback.

---

### Official Review · Reviewer_x5z4 · 2025-07-03

**Clarity:** 3
**Significance:** 3
**Originality:** 2
**Rating:** 5
**Confidence:** 3

**Summary:**

The paper uses influence functions to analyze which SFT examples cause reasoning abilities, focusing on math and coding.  It finds that difficult math examples improve both math and code reasoning, whereas easy code examples improves code reasoning.  This insight is used to re-construct the SFT dataset, to improve AIME and LiveCodeBench performance.  Furthermore, token-level attribution also reveals that "exploration" behavior stimulates general reasoning.

**Questions:**

- what was the reason that token-level attribution used gradients, whereas sequence-level attribution used ablation, and not just the sum of token-level attributions from gradients?
- was normalizing the influence scores so that magnitude of the gradient is ignored, i.e. RelatIF, considered when doing the analysis?  I wonder if sequence length would correlate with influence, if there is not normalization.

**Ethical Concerns:**

["NO or VERY MINOR ethics concerns only"]

**Final Justification:**

Clarification that novelty is not in methods development, but application of existing methods, which other reviewers noted is of practical importance.

**Quality:**

3

**Strengths And Weaknesses:**

Strengths
- The analysis is important in that it fills a gap that needed to be filled, and at the example-level, confirming previous heuristics for constructing reasoning SFT data.
- The paper does token-level analysis, which provides deeper qualitative insights to guide dataset construction, and at the sequence-level, finding counter-intuitive insight that "exploration" behavior can improve reasoning.

Weaknesses
- Method-wise, my understanding is that there was not significant novel innovation - the Anthropic influence functions paper already proposed token-level influence.

---

> ### Author Rebuttal · Authors · 2025-07-31
>
> Thank you for the positive feedback and useful suggestions! We are glad you think our analysis is important in that it fills a gap that needed to be filled. We address your concerns point by point below.
>
> **Weakness:**  Method-wise, my understanding is that there was not significant novel innovation.
>
> Thank you for your feedback. We would like to emphasize that **our main contribution lies in the novel application of influence functions to trace how training data during the SFT phase shapes LLM reasoning for the first time**, presenting key findings with substantial practical value, as acknowledged by Reviewer mqmh. Reviewer AwiZ also notes that we 'target an important problem and uncover multiple interesting findings'.
>
> **Q1:** What was the reason that token-level attribution used gradients, whereas sequence-level attribution used ablation, and not just the sum of token-level attributions from gradients?
>
> **Summing token-level attributions from gradients only treats the sequence in its role as a prediction target and ignores its function as input context** (see Equation 7) due to the AR nature. In contrast, sequence‑level attribution via ablation quantifies the effect of removing the sequence from training data, thereby capturing its contribution both as input and as target. Thank you for this question, and we will add this clarification to the revised version.
>
> **Q2:** Was normalizing the influence scores so that magnitude of the gradient is ignored, i.e. RelatIF, considered when doing the analysis? I wonder if sequence length would correlate with influence, if there is not normalization.
>
> Thank you for raising this interesting question. First, we would like to clarify that influence scores, as defined in Equation  3, measure the effect on the model performance when a training sample is removed; accordingly, we do not normalize by gradient norm or sequence length, since the raw gradient magnitude itself may carry information about a sample’s impact.
>
> Furthermore, to investigate whether sequence length would correlate with influence, we compute the Spearman rank correlation between the original influence scores and those obtained after dividing by sequence length of 5000 samples, yielding ρ = 0.48. This indicates moderate correlation, which is consistent with findings in related works[1,2].
>
> [1] The Impact of Reasoning Step Length on Large Language Models. https://arxiv.org/abs/2401.04925
>
> [2] LIMO: Less is More for Reasoning. https://arxiv.org/abs/2502.03387
>
> We hope these reclarifications and explanations of our method in response to your initial concerns can convince you to increase your score.

---

> > ### Comment · Reviewer_x5z4 · 2025-08-08
> >
> > Thank you for the clarifications.  Looking at the other reviews, I acknowledge the practical value of the paper, even if the innovation in terms of methods development is limited, and raise my score.  Also, I appreciate the references to point out that sequence length is expected to have an effect on reasoning, so that gradients should not be normalized.

---

> > > ### Author Response · Authors · 2025-08-09
> > >
> > > We appreciate your kind consideration of our response and the improved score. Thank you again for your valuable feedback!

---

### Decision · Program_Chairs · 2025-09-17

**Decision:**

Accept (poster)

**Comment:**

This paper examines how SFT data influences reasoning in LLMs, using influence functions to study both math and coding tasks. The work provides several interesting findings, such as the benefits of difficult math problems for cross-domain reasoning and the positive impact of exploratory reasoning traces. Reviewers appreciated the clear motivation, practical relevance, and the potential to guide dataset design. Concerns were raised about limited methodological novelty, missing comparisons to alternative data selection strategies, and the absence of stronger statistical validation. The rebuttal helped clarify some points and alleviated part of these concerns, though not all. Overall, despite some limitations, the paper offers timely insights into an important problem and I recommend acceptance.